# The Role of Gut Dysbiosis in the Pathophysiology of Neuropsychiatric Disorders

**DOI:** 10.3390/cells12010054

**Published:** 2022-12-23

**Authors:** Nikhilesh Anand, Vasavi Rakesh Gorantla, Saravana Babu Chidambaram

**Affiliations:** 1Department of Pharmacology, American University of Antigua College of Medicine, University Park, Jabberwock Beach Road, Coolidge, Antigua and Barbuda; 2Department of Anatomical Sciences, St. George’s University School of Medicine, St. George’s University, Saint George, Grenada; 3Department of Pharmacology, JSS College of Pharmacy, JSS Academy of Higher Education & Research, Mysuru 570015, Karnataka, India; 4Centre for Experimental Pharmacology & Toxicology, JSS College of Pharmacy, JSS Academy of Higher Education & Research, Mysuru 570015, Karnataka, India

**Keywords:** gut microbiota, gut dysbiosis, microbiota gut–brain axis, neuropsychiatric disorders, inflammation, oxidative stress

## Abstract

Mounting evidence shows that the complex gut microbial ecosystem in the human gastrointestinal (GI) tract regulates the physiology of the central nervous system (CNS) via microbiota and the gut–brain (MGB) axis. The GI microbial ecosystem communicates with the brain through the neuroendocrine, immune, and autonomic nervous systems. Recent studies have bolstered the involvement of dysfunctional MGB axis signaling in the pathophysiology of several neurodegenerative, neurodevelopmental, and neuropsychiatric disorders (NPDs). Several investigations on the dynamic microbial system and genetic–environmental interactions with the gut microbiota (GM) have shown that changes in the composition, diversity and/or functions of gut microbes (termed “gut dysbiosis” (GD)) affect neuropsychiatric health by inducing alterations in the signaling pathways of the MGB axis. Interestingly, both preclinical and clinical evidence shows a positive correlation between GD and the pathogenesis and progression of NPDs. Long-term GD leads to overstimulation of hypothalamic–pituitary–adrenal (HPA) axis and the neuroimmune system, along with altered neurotransmitter levels, resulting in dysfunctional signal transduction, inflammation, increased oxidative stress (OS), mitochondrial dysfunction, and neuronal death. Further studies on the MGB axis have highlighted the significance of GM in the development of brain regions specific to stress-related behaviors, including depression and anxiety, and the immune system in the early life. GD-mediated deregulation of the MGB axis imbalances host homeostasis significantly by disrupting the integrity of the intestinal and blood–brain barrier (BBB), mucus secretion, and gut immune and brain immune functions. This review collates evidence on the potential interaction between GD and NPDs from preclinical and clinical data. Additionally, we summarize the use of non-therapeutic modulators such as pro-, pre-, syn- and post-biotics, and specific diets or fecal microbiota transplantation (FMT), which are promising targets for the management of NPDs.

## 1. Introduction

Trillions of microorganisms colonizing the skin, as well as the nasal, oral, pulmonary, GI and vaginal mucosal cavities, have been found to be crucial regulators of human metabolic, immune and GI homeostasis [1]. The enteric microbiota, or gut microbiota (GM), refers to the symbiotic co-evolution of the majority of microorganisms (including bacteria, fungi, viruses, and protozoa [2]) that colonize the human GI tract [3]. The GM interacts with multiple organs in the host to form a multiplex of essential pathways that govern homeostasis. Although gut microbial ecology is unique for each individual, there appears to be a certain balance in the composition and diversity that benefits the host [4,5]. Gut microbes regulate digestive, immune, endocrine, and neurological functions via a highly interconnected host–microbiome system [6,7]. Gut microbes communicate with the brain either directly or indirectly in a complex and multidimensional manner via neural, immune, and entero-endocrine signaling pathways [8], forming the MGB axis. Moreover, the brain signals the gut (top-to-bottom approach) by regulating the sensory, motor, and secretory modes of the GI tract [9,10], while the gut signals the brain (bottom-to-top approach) by modulating higher cognitive and behavioral functions [11,12]. Generally, gut dysbiosis (GD) or gut microbial dysbiosis refers to an imbalance in the diversity and composition of GM. Several experimental animal models and cross-sectional clinical reports provide evidence that GD is associated with a wide variety of GI, metabolic, cerebrovascular and CNS-related diseases [13,14,15,16]. Recent technical developments in the field of neuroscience and neuroimmunology have shown that GD is closely related with the etiopathology and pathophysiology of NPDs. In support of this, several clinical studies have evidently proven the presence of GI dysfunctions in patients with anxiety, depressive, and autistic disorders [17,18]. Patients with NPDs frequently report GI symptoms such as altered bowel habits, constipation or diarrhea, chronic abdominal pain, nausea, vomiting, and colic [19,20].

Specifically, the pathogenic involvement of GD in a plethora of NPDs, including stress-induced disorders (depression, anxiety and major depressive disorders (MDD)), psychiatric disorders (schizophrenia (SZ) and bipolar disorders (BD)), and neurodevelopmental disorders (attention deficit hyperactivity disorders (ADHD) and Asperger’s syndrome or autism spectrum disorders (ASD)) [14,21,22,23] are elucidated in this review. This review summarizes the prudential role of molecular pathogenic mechanisms involved in GD-associated NPDs, and also discusses the potential therapeutic approaches using pro-, pre-, and syn-biotics, diets, and nutrition as well as fecal microbial transplantation (FMT).

### 1.1. Gut Microbiota

Approximately 10^13^–10^18^ different microorganisms, including bacteria (10^14^, mostly anaerobic), archaea, fungi, and viruses, form the human microbiota [1,24]. The term ‘microbiota’ refers to the composition, density and diversity of the microorganisms, while the term ‘microbiome’ refers to the genetic and functional characteristics of the GM. The adult intestinal microbiome comprises more unique genes (about 150 times) than the entire human genome [25]. More than 200,000 to 1,000,000 bacterial genes are represented by gut microbes [24]. Each human has at least 160 species of bacteria, and more than 3 million microbial genes [26]. The total number of bacteria in a human weighing 70 kg is estimated to be 3.8 × 10^13^, which is greater than the total number of human cells (approximately 3.0 × 10^13^) [1]. The adult microbiome weighs approximately the same as the human brain [6]. The main enterotypes found in the human GM are Prevotella, Bacteroides, and Ruminococcus genera [26].

A typical GM comprises four major phyla, Firmicutes, Bacteroidetes, Proteobacteria and Actinobacteria, and two minor phyla, Fusobacteria and Verrucomicrobia [27,28]. Both the composition and the diversity of the GM in infants are determined by endogenous factors such as the host genome, health status, mode of delivery (natural vaginal delivery or caesarean section) and feeding (breast fed or formula fed) [29,30], and are further determined by exogenous factors such as long-term dietary habits, infection, use of antibiotics, lifestyle, physical activity, and stress exposure in adults [31,32,33]. Even though the gut microbial composition is similar in terms of relative composition and distribution among healthy people over time [34], microbial composition is quite stable and unique to each person, and can be considered a personal microbial signature or enterotype fingerprint [35]. After 20 years of promising results on MGB axis research, the GM is now being referred to as a new organ [36]. These findings emphasize the importance of gut microbes in maintaining homeostasis and optimal functioning.

### 1.2. Gut-Microbial-Derived Metabolites

The GM ferment undigested complex dietary fibers and produce essential bioactive metabolic products, known as gut-derived metabolites, such as short-chain fatty acids (SCFAs), long-chain fatty acids, branched-chain amino acids, trimethylamine-N-oxide (TMAO), lipopolysaccharide (LPS), bile acids, and catecholamines [37,38,39,40]. Gut microbes also regulate the secretion of host-derived vitamins (B12), nitric oxide (NO) and indoles, and synthesize and/or induce the production of neurotransmitters including tryptophan, 5-HT, glutamate, γ-aminobutyric acid (GABA), acetylcholine, dopamine, histamine, and noradrenaline [18,41,42,43,44,45]. Furthermore, microbes in the GI tract regulate the release of various gut hormones that improve glucose homeostasis and behavioral changes, such as peptide YY (PYY); regulate the physiology of immune cells, anxiety and mood, such as neuropeptide Y; and are involved in the regulation of hunger and satiety, such as leptin, ghrelin, cholecystokinin, and glucagon-like peptide-1 [46,47]. The secretion of several metabolites occurs either in a direct (secreted by the microbes themselves) or indirect manner (secreted from other cells stimulated by microbes) by interacting with the CNS and enteric nervous system (ENS). Anaerobic fermentation of macronutrients (such as complex dietary carbohydrates) by microorganisms induces the secretion of SCFAs such as propionate, acetate, and butyrate. In the hypothalamus, levels of neurotransmitters such as glutamine, glutamate and GABA are mainly modulated by SCFAs. The majority of serotonin production happens in the gut as a result of SCFA-mediated upregulation of tryptophan 5-hydroxylase 1, a key enzyme involved in 5-HT synthesis. SCFA also promotes the expression of tyrosine hydroxylase (an important regulatory enzyme involved in adrenaline, dopamine and noradrenaline formation) [48,49]. These results highlight the complex physiological role of gut microbes and their metabolites on the maintenance of nervous system homeostasis (central or endocrine or autonomic), either directly or indirectly.

### 1.3. Microbiota Gut–Brain Axis

Research on the signaling pathways involved in the MGB axis and its regulatory functions is still being explored [50,51]. The complex communication between the gut and host brain occurs mainly through the extrinsic nerves of the GI tract, which stimulate the vagal and spinal afferent fibers and signals the brain, while the brain reciprocally signals the gut using efferent sympathetic and parasympathetic fibers [17,27,52]. Indeed, there are direct and indirect pathways through which the GM affect brain function and vice versa [9,47,53,54]. Gut microorganisms can affect higher CNS functions in a direct way via the highly complex MGB axis by means of:▪The neural network (such as vagus nerve by producing bacterial metabolites, the intrinsic branches of the ENS, and the extrinsic parasympathetic and sympathetic branches of the autonomic nervous system) [9,43].▪The endocrine system (via HPA axis) [55,56].▪The immune system (by producing cytokines and chemokines from both peripheral and CNS infiltrating immune cells) [57].▪Barriers such as the BBB and the gut mucosal barrier [58,59,60].

On the other hand, gut microbes can indirectly influence the CNS via the production of metabolites like SCFAs and neural mediators or inducing the release of 5-HT from EECs [18,38,39,40].

Conversely, the brain signals the gut mainly via the HPA axis in the presence of stressful stimuli. Importantly, the HPA axis provides the fundamental response to the environmental, social and physiological stressors by hyperactivating the immune cells to release proinflammatory cytokines. Increased levels of cortisol increase the permeability of the intestinal barrier, which leads to leaky gut and increased LPS levels (released from the Gram-negative bacterial cell wall and commonly termed as ‘endotoxemia’), eventually resulting in GD [61,62,63,64]. Excessive release of LPS into the blood stream induces systemic and neuroinflammation via HPA hyperactivation, leading to elevated secretion of cortisol, which correlates positively with depression [65]. Thus, MGB axis signaling pathways exert a significant influence on CNS homeostatic processes, such as neurotransmission, neurogenesis, activation of the stress axes, and neuroinflammation [6,30,47,66,67,68,69], in addition to modulating complex behaviors, such as sociability and anxiety [43,70]. The MGB axis acts as a major highway connecting the gut with the brain and vice versa. Therefore, aberrations in either gut or brain affect the homeostasis of the other, provoking a vicious inflammatory cascade in the peripheral nervous system and CNS.

### 1.4. Gut Dysbiosis

Specifically, GD can be defined as an increased number of pathogenic microbes or pathobionts (microbes producing pro-inflammatory cytokines) and decreased number of beneficial microbes or symbionts (microbes producing anti-inflammatory cytokines). The pathogenic representation of GD-induced NPDs [71,72] includes
▪Altered microbial composition and diversity and metabolites, which in turn alter the levels and/or synthesis of neurotransmitters (5-HT, dopamine, noradrenaline, and glutamate), leading to deregulated MGB signaling.▪Reduction in the number of goblet cells leads to a decrease in mucus production and mucosal layer becomes thin.▪Decreased expression of tight junction proteins (including claudin-5 and occludin) between the intestinal epithelial cells leading to increased permeability, resulting in leaky gut (also known as leaky gut syndrome) [11,73].▪Increased translocation of pathobionts and its toxic components such as LPS and peptidoglycan into the systemic circulation leading increased secretion of pro-inflammatory cytokines (like interleukin (IL)-18, IL-1, IL-6, and TNF-α).▪Leaky gut evokes chronic systemic inflammation by breaking down the integrity of the BBB by disrupting the tight and anchoring junction proteins in the frontal cortical, hippocampal, and striatal regions, which eventually alters the brain functions.▪Increased BBB permeability results in excessive translocation of immune cells and toxic microbial metabolites into the brain, which in turn enhance cytokines, chemokines, and endocrine (stress) messengers in the brain parenchyma.▪Altered neuroimmune status is marked by alterations in microglial maturation, neurogenesis, myelination, neurotrophin expression, neurotransmitters, and their respective receptors [18,60,74,75].

Collectively, these findings indicate that GD deregulates MGB signaling and begins the pro-inflammatory cascade, which correlates directly with the development of symptoms of NPDs [76,77].

### 1.5. Search Methods and Selection Criteria

Database sources include SCOPUS, MEDLINE, PubMed, Cochrane, PsycINFO, Nature, and ScienceDirect. The relevant randomized controlled trials from the Journals of neuropsychiatry, neuroimmunology, neuroscience, neurogasteroenterology, immunology and microbiology were hand searched and selected. In addition, recent articles investigating the correlation of gut microbial alterations and neuropsychiatric disorders were selected, and citation searches of selected articles were performed.

## 2. Pathogenic Link between Gut Dysbiosis and Neuropsychiatric Disorders

### 2.1. Immune-Mediated Inflammatory Response

The pathogenic link between GD and NPDs on disease onset and progression results from the abnormal physiology of the immune system. During the early stages of human development, the dynamics, composition and distribution of GM is regulated by the immune system [78], in turn, microbes influence the maturation and function of immune system [79].

Commensal GM are actively involved in priming the innate and/or adaptive immune responses [80,81]. The direct and indirect interactions of GM and their metabolites with various cellular components in the CNS occurs via the stimulation of immune signaling pathways like inflammasome, type 1 interferon and nuclear factor kappa-light-chain-enhancer of activated B cells (NF-κB) signaling pathways and promote the subsets of CD4+ Th cells [82,83]. GD affects the brain functions either directly or indirectly via the immune system, and leads to the development of NPDs. On the other hand, stress exposure (acute or chronic) can induce GD via a dysfunctional MGB axis, in turn provoking the onset of or aggravating existing neuropsychiatric conditions [43,84] by inducing systemic and neuronal inflammation.

Exposure to stress leads to increased production of HPA axis hormones with elevated levels of corticotrophin releasing factor, adrenocorticotropic hormone (ACTH), corticosterone/cortisol levels and a reduced level of glucocorticoid receptor expression in GF mice [85]. These changes directly affect the intestinal barrier integrity, GI motility and mucus secretion alterations in the GM composition [66]. Then, increased gut permeability, termed as ‘leaky gut syndrome’, leads to excessive translocation of Gram-negative bacteria and their products, such as LPS and peptidoglycan [86]. LPS and peptidoglycan found in the intestinal microbes are typical examples of inflammation-inducing substances that can be recognized as pathogen-associated molecular patterns or damage-associated molecular patterns by Toll-like receptors in the innate immune cells such as CD4+ T helper (Th) cell subsets, myeloid cells and mast cells [87,88], which leads to the development of chronic systemic low-grade inflammation. Activated immune cells release increased levels of pro-inflammatory cytokines (IL-6, TNF-α and IL1β) and several chemokines (monocyte chemoattractant protein-1, CXCL-1, and MIP 1α), which leads to hyperactivation of HPA axis and increased cortisol secretion [89,90]. The indirect effects of intestinal microbes include modified levels of neurotransmitter precursors in the gut lumen along with modulation in the synthesis of neurotransmitters like GABA, 5-HT, dopamine (DA), and noradrenaline (NA) [18,43,45]. Some studies have shown that repeated administration of LPS during adolescence or early adulthood produces anxiety or depressive-like or sickness behavior such as fatigue, anorexia, low mood, or apathy later in life, which are considered important risk factors for the development of NPDs [91,92].

Clinically, activated inflammasome and elevated levels of proinflammatory cytokines (IL-1β, IL-6, and IL-18) have been noted in patients with MDD. Campylobacter jejuni infection induces GD by overstimulating NF-κB signaling pathways, leading to excessive release of various cytokines and activation of immune cells [93]. Experimental animal studies have shown that MGB deregulation in early life is correlated with GD, reduced brain-derived neurotrophic factor (BDNF), increased HPA axis activation, impaired glucocorticoid receptor-mediated negative feedback, increased stress reactivity, aberrated brain development, and abnormal behavior such as impaired social interaction, anxious-like, cognitive deficit, and other metabolic, immune or psychological disorders in adulthood [94,95,96]. These results indicate that stress-induced GD affects brain and immune functions, especially during the perinatal period, early childhood, and adolescence, which are reflected in adulthood [57,97] (Figure 1).

### 2.2. Microglial Dysfunction and Impaired Neural Circuitry

Microglia are the important sentinel immune cells in the CNS. Aberrations of normal microglial functions such as synaptic pruning, phagocytosis of cellular debris, and the release of cell signaling factors such as neurotrophins, proinflammatory cytokines, and extracellular matrix components, which results in the abnormal brain connections and brain volumes, are found more commonly in brain samples of NPDs [77,98,99,100]. In the synaptic pruning process, the vital functions of microglia include removal of the infrequently used synapses, and regulation of the synapse organization and synapse assembly in the brain throughout the lifespan [101,102,103,104,105]. Mounting evidence indicates that certain brain areas of subjects with NPDs have reduced density (as found in MDD patients) or elevated density (as noted in SZ patients) of microglial cells than nonpsychiatric controls [106]. Several studies have shown that GM have been shown to influence the maturation progress of naïve microglia [107]. For example, microglia in GF mice displayed immature phenotype and aberrated gene expression profiles [74]. From the prenatal period, microglial cells sex- and time-dependently respond to changes in microbiota composition and its metabolites [108]. Another study showed that antibiotic-treated or GF adult mice exhibit significant decline in fear extinction learning and confirmed the presence of immature microglia with an altered genetic profile [109]. In humans, GM can directly regulate the microglial activation and function during sensitive stages of brain development [74], which in turn can shape the neuronal circuits by altering synaptic pruning [110]. It is important to note that both synaptic pruning and gut microbial maturation occurs at the same time. Conclusively, GD can indirectly result in the formation of defective neural circuit, which causes behavioral abnormalities, mainly by altering microglia-mediated synaptic pruning and dendritic spine remodeling. Clinical studies have supported these findings, as deregulation of microglial activity were found to correlate positively with various NPDs including SZ [111,112,113], BD [114,115,116], ASD [117,118,119] and ADHD symptoms [99,120,121]. Surprisingly, in ASD and ADHD, onset of symptoms begins simultaneously with the initiation of the synaptic pruning process during early life [122,123,124,125]. On the other hand, the onset of symptoms occurs in parallel with the terminal stages of the synaptic pruning process in SZ and BD [116,126,127].

Taken together, ASD children have an increased brain volume due to excessive presence of neurons and synaptic networks in the brain cortex [128,129,130,131]. Brain samples of ASD patients showed increased expression of genes related to microglia functions when compared to controls [132], along with an increase in glial markers and a decrease in the expression of synaptic genes, compared to healthy individuals [133]. In transgenic mice with a targeted deletion of autophagy-related gene 7 (atg7), microglia-mediated abolishment of synaptic pruning led to increased number of immature dendritic spines and defective social behavior circuit, concurrent to ASD pathogenesis [134]. The cerebral cortex, white matter, and cerebellum of ASD patients displayed increased number of activated microglia [135]. In contrast, a reduced number of inactivated microglia were found to be shown in the grey and white matters, but increased numbers of activated microglia were found in the grey matter of ASD post mortem brain tissues [119]. SZ patients showed grey matter reduction due to enhanced synaptic pruning mediated by microglia in the cortex and thalamus [136]. SZ patients have fewer synapses due to excessive synaptic pruning and suboptimal fine-tuning of neural circuits which accounts for abnormal motor, sensory and cognitive behavior [137,138,139,140]. A recent meta-analysis on post mortem brain studies showed significantly increased microglial density (especially in temporal cortical areas), and increased expression of pro-inflammatory genes and microglial markers [141], confirming the immune involvement in SZ pathogenesis.

These findings highlight that stress-induced GD in different developmental stages of life (infancy, childhood or adulthood) may exert behavioral alterations in adulthood by hyperactivating the HPA response, redesigning the neural circuitry and affecting the maturation and functions of CNS immune cells.

## 3. Preclinical Evidence on Pathogenic Link between Gut Dysbiosis and Neuropsychiatric Disorders

Numerous studies on animal models have confirmed the link between GD and the onset and/or progression of NPDs. To investigate how GM regulate brain development and function, researchers developed animals completely lacking microbiota, referred to as germ-free (GF) mice, and treated animals with antibiotics or animals colonized with complex microbiota except particular pathogenic flora (specific pathogen free (SPF)). Also, the interlink between gut microbiome and chronic unpredictable mild stress (CUMS) induced depression is well established (Figure 2). 

### 3.1. Experimental Studies in GF Mice

GF mice have deficiencies in spatial learning, working memory, recognition, and emotional behaviors, indicating neural dysfunction [17,142]. GF mice display impaired social behavior, hyperactivity and lower anxiety [142,143,144,145] vs. normally colonized (wild-type (WT)) mice. In GF mice, abnormal behavior and psychological symptoms were reasoned by the alterations in the expression levels of several neurotransmitters and related receptors such as 5-HT, GABA, noradrenaline and dopamine in specific brain areas [143,146]. Disruption of gut microbes during early adolescence leads to a significant decrease in oxytocin expression in the adult brain [97]. GF mice showed altered levels of brain substrates such as corticosterone, 5-HT, BDNF and pro-inflammatory markers known to influence adult hippocampal neurogenesis [147].

GF mice have higher tryptophan and lower 5-HT levels in blood compared to WT mice [148]. Additionally, GF mice have high levels of stress hormone and low levels of BDNF [17,55]. GF mice have impaired hippocampal morphology with increased neurogenesis in dorsal hippocampus, and immature microglia as well as significantly altered BBB permeability and different levels of serotonin, adrenaline, noradrenaline, and dopamine [60,145,149,150]. GM regulate the structural and functional changes of the amygdala (a region associated with social and fear-related behaviors) and prefrontal cortical myelination, thus alterations in these regions and their functions are commonly found in patients with NPDs and animals models mimicking psychiatric conditions [151,152]. Colonization with complex healthy microbiota reverses the intrinsic and extrinsic nerve function and MGB signaling in GF mice [153] and regulates the dysregulated microRNA expression in prefrontal cortical regions and amygdala of GF mice [154]. Overall findings of these studies postulate the absence of gut microbes affects the higher brain functions such as social behavior, cognitive skills and fear conditioning, as a result of alterations in the levels of neurotransmitters, neural growth factors, and stress hormones, as well as abnormal changes in neural synaptic process, pruning, neurogenesis and neural circuitry.

### 3.2. Experimental Studies in SPF Mice

GF mice showed enhanced exploratory and reduced anxiety-like behavior than SPF mice in the standard behavioral tests [143,155]. Heijtz et al. reported increased expression of neurotrophins (including nerve growth factor and BDNF), and different expression of genes involved in the secondary messenger signaling pathways and long-term synaptic processes in the hippocampal, cortical and striatal regions of SPF mice compared to that of GF mice. Indeed, the expression of N-methyl-d-aspartate receptor (NMDA) receptor subunit (NR2B) and 5-HT receptor 1A were also increased in the central amygdala and hippocampus of SPF mice in comparison to GF mice [143]. These findings indicate the direct role of specific gut microbes on certain signaling pathways and associated alterations in particular brain regions. These studies enable the researchers to understand specific links between particular brain regions involved in the pathophysiology of the specific NPDs, and thus targeting those microbial population improve the clinical symptoms restoring the underlying synaptic aberrations.

### 3.3. Experimental Studies in Mice Treated with Antibiotics

SPF mice receiving cocktail of neomycin, bacitracin and pimaricin showed GD to be accompanied by increased exploratory behavior, reduced anxiety, and altered BDNF levels in the hippocampus and amygdala [157]. SPF mice (infected with Citrobacter rodentium) showed memory dysfunction upon exposure to acute stress, while GF mice showed defective memory and cognition with or without exposure to stress [142]. Antibiotic-induced GD affects the cognitive performance in novel object recognition tests, which is associated with BDNF, NR2B, 5-HT transporter, and neuropeptide Y [158]. Several studies using broad-spectrum antibiotic cocktails have produced results similar to those of GF mice, including impaired social behaviors, neurogenesis and cognitive function together with GD. Similar to animal microbiota models (SPF or GF mice), treatment with antibiotics replicated similar findings, confirming the key functioning of microbes in cognition and behavior.

Taken together, preclinical studies dictate the physiological role of gut microbes and its metabolites in optimum functioning of immune, metabolic and neural systems. GD provokes dysfunctions in these systems stemming from pathological changes such as increased inflammatory cascade, aberrated levels of neurotransmitters or neural growth factors (e.g., BDNF), synaptic processes, neuronal death, or volumetric changes. Taken together, these pathological changes translate into psychological symptoms such as increased or reduced anxiety, increased stress or fearful response, altered social behavior, cognitive decline or memory deficit. Specifically targeted deletion of microbial species dictates their specific role in these regulatory functions. These results clearly indicate the importance of MGB axis and healthy gut microbial composition in the regulation of brain functions.

## 4. Clinical Evidence on a Pathogenic Link between GD and Neuropsychiatric Disorders

Much evidence from clinical data strongly suggests a positive correlation between GD and the development and/or manifestation of different neuropsychiatric conditions including depression, anxiety, schizophrenia, ASD, and ADHD. High-throughput genetic sequencing and metabolomics also reported alterations in GI microbiota and fecal metabolic phenotype related to depressive disorders [159,160]. The common NPDs and their correlation with GD are discussed below.

### 4.1. Stress-Related Disorders

Stress is a physiological response to environmental or psychological challenges. Different types of stress such as acute/chronic, mild/severe, only once/monotonous [161] due to environmental, biological, and psychosocial factors have shown to elicit various responses unique to each individual. Indeed, acute stress exposure alter GM composition and increase the vulnerability to develop risk factors for NPDs [162], while chronic stress exposure leads to leaky gut syndrome, increasing the inflammatory response, which can be functionally related to the onset of NPDs [66]. Additionally, stress exposure in different stages of life like early pre-, post-natal period/infancy or adulthood shape the gut microbiome community, which programs the stress responsiveness for the rest of the life [163,164,165,166]. An exploratory study revealed that patients with severe forms of post-traumatic stress disorder have reduced relative abundances of Actinobacteria, Verrucomicrobia, and Lentisphaerae compared to trauma-exposed controls [95]. A novel arousal-based individual screening model [167], susceptible mice (exhibiting long-lasting hyperarousal after 24-h restraint) displayed chronic PTSD-like phenotypes such as exaggerated fear reactivity, avoidance of trauma-related cue, increased avoidance-like behavior and social/cognitive impairment. These findings correlate with the presence of altered transcription of PTSD-related genes, HPA dysfunction and impaired hippocampal synaptic plasticity in susceptible mice. Similarly, a study on a mouse model of post-traumatic stress disorder developed by aggressor-exposed social stress (mimicking warzone-like conflicts) showed marked alterations in the relative abundance of time-resolved ratios of *Firmicutes* and *Bacteroidetes* as well as *Verrucomicrobia* and *Actinobacteria* [168]. Collectively, the stimulation of the sympathetic nervous system and defective gut barrier integrity combined with changes in gut secretion and motility define the pathogenic mechanisms induced by the acute stress, while the chronic stress disturbs the homeostatic connection between the GM and the host, resulting in inflammatory responses, which directly affects brain functions. Clinical evidence supporting the pathological role of GD in the development and maintenance of stress-related disorders makes the MGB axis a promising therapeutic target.

### 4.2. Anxiety

Anxiety and depression are the most commonly reported mood disorders. Anxiety is characterized by frequent nervous behavior, rumination, fidgeting, worried thoughts, negative thinking, and restlessness or agitation, causing significant functional impairment in life activities. The pathogenesis involves abnormal changes in nervous, endocrinal, and immunological systems [169]. Exposure to stress (including environmental, biological, or psychological) can trigger or aggravate anxiety responses by stimulating HPA axis or immune response [170]. Several clinical studies have documented the co-presence of anxiety and gastric symptoms due to GD [20] accompanied by altered levels of neurotransmitters and immunologic factors via dysfunctional MGB [9,171]. A longitudinal pilot study showed a marked reduction in the both richness and diversity of microbial population, characterized by discrete metagenomic composition such as lower abundance of SCFAs-producing bacteria and higher abundance of Escherichia-Shigella, Fusobacterium and *Ruminococcus gnavus* [172]. GD caused by bacterial infection can aggravate the anxiety through immunologic and metabolic pathways of the MGB. For instance, Campylobacter jejuni infection increases depressive- or anxiety-like behaviors by activating c-Fos proteins (markers of neuronal activation) [173], while infection by Citrobacter rodentium increase anxiety presumably via the vagal sensory neurons [174]. However, Trichuris muris infection increased anxiety via both immunologic and metabolic mechanisms [157].

### 4.3. Depression

The most common form of mood disorder in the digital epoch, found in millions, is depression, which is characterized by anhedonia, lack of motivation, low state of mood, hopelessness, morbidity of depression and anxiety affects important functioning of life areas, such as low self-esteem, severe fatigue, frustration, intrusive thoughts, and restlessness [175]. Co-academics, work, relationships and other things have led to increased disability around the globe (World Health, 2017). Several factors, like stress, poor diet, low or no physical activity, obesity, low-grade inflammation, smoking, atopy, low sleep hygiene, dental care, and nutritional deficits like vitamin deficiency (mainly vitamin D) contribute to the development of depression [176,177]. There is common speculation that neuroimmunological dysregulation can lead to development of depression [169,178]. Importantly, depression is associated with hyperactivation or deregulation of HPA axis, dysregulation of the neuroimmunological and neurotransmitter signaling pathways, and deficiency of tryptophan metabolism [169,178,179].

Several observational and clinical research studies have proved the direct correlation of depression with GD, confirming its role in depression pathology [66,180,181]. GD itself can also induce dysregulation of inflammatory, stress (HPA) or neurotransmitter signaling pathways [20,67,153,182], leading to onset of depression. For instance, GD associated with impaired GI barrier or leaky gut allows Gram-negative bacteria (Enterobacteriaceae) to enter systemic circulation, activating immunoglobulin (IgA and IgM)-mediated immune responses to peptidoglycan component (LPS) of E. coli leading to specific inflammatory pathways [183,184] via secreting endotoxins, which eventually results in depression [176]. Furthermore, FMT from patients with alcoholism to antibiotic (ABX)-depleted microbiota model of C57BL/6J mice was shown to induce anxiety/depression-like phenotype and reduce social interaction. FMT-treated GM-depleted mice showed reduced expression of BDNF, α1 subunit of GABA type A receptor in prefrontal cortex, and metabotropic glutamate receptors 1/protein kinase C ε levels in nucleus accumbens [100]. Conversely, colonization of chronically alcohol-exposed animals with fecal microbiota from healthy donors showed reduction in anxiety and depression-like phenotype [185]. Interestingly, colonization of antibiotic-induced pseudo-GF mice with fecal microbiota from anhedonia-susceptible rats markedly increased the anhedonia, pain and depressive symptoms compared to colonization with fecal microbiota from resilient rats [160]. Clinical studies report that depressed patients display GD indicated by reduced bacterial diversity and species richness [66,186]. Jiang et al. also report that altered GM composition correlates negatively with MDD [172]. A clinical study revealed the presence of GD in patients with MDD when compared to non-depressed volunteers [185].

A recent cross-sectional study in post partum women with severe and suicidal depression indicated the presence of higher levels of IL-6 and IL-8 and reduced concentrations of IL-2, 5-HT, and quinolinic acid in plasma due to dysregulation of kynurenine pathway [187]. MDD patients exhibit higher levels of serum antibodies due to the excessive presence of LPS from gram-negative enterobacteria than those in controls [180]. Animal models receiving FMT from MDD patients showed stress-induced increased intestinal permeability and microbial translocation [188,189]. Moreover, FMT from depressed subjects to GF mice induced depression and anxiety-like phenotype [164,185,190]. Six major studies investigating the microbial composition in MDD patients indicated the higher relative abundance of *Blautia Klebsiella*, *Anaerostipes*, *Clostridium, Parasutterella, Parabacteroides, Phascolarctobacterium*, *Lachnospiraceae incertae sedis*, and *Streptococcus; lower relative abundance of Dialister*, *Faecalibacterium*, *Bifidobacterium, Ruminococcus*, and *Escherichia/Shigella*; and altered composition of *Alistipes, Megamonas*, *Bacteroides*, *Roseburia*, *Prevotella*, and *Oscillibacter* [72,191,192].

### 4.4. Autism Spectrum Disorders

Autism spectrum disorders (ASD) are defined as a progressive developmental disabilities found commonly in children and adolescents, which are characterized by impaired social behavior and communication, along with presence of limited, repetitive and stereotyped interests and behavior. In utero valproate animal model of autism show an increase in the Firmicutes/Bacteroidetes ratio as well as increases in cecal butyrate levels and genera *Alistipes, Mollicutes*, *Lactobacillales*, and *Enterorhabdus* [193]. Almost half of children with ASD frequently report GI symptoms such as altered bowel habits, constipation, colic, gastroesophageal reflux, diarrhea, bloating, and chronic abdominal pain [194]. Moreover, GI symptoms of ASD patients seem to correlate strongly with the severity of their behavioral and emotional symptoms like irritability, aggressiveness, temper tantrums, and sleep problems [19,195,196]. Gastric symptoms are reported frequently in ASD patients, which is believed to related to neurologic cause rather than a gastroenteric cause [197], and chronic gastric disturbances is considered to be a risk factor for ASD development [195,198].

In ASD patients, GD is represented by mucosal inflammation indicated by the increased presence of inflammatory markers (IL 6, IL 1β, TNF and monocyte chemotactic protein 1) in cerebrospinal fluid [197,199,200] and immune dysfunction in the GI tract indicated by the infiltration of immune cells such as CD3+ TNFα+ cells or CD3+ IFNγ+ cells, monocytes, and natural killer cells which produces pro-inflammatory cytokines [135,201]. Leaky gut syndrome resulting in higher serum LPS levels was found to be correlated with deficits in social skills in autistic patients when compared to healthy controls [202], indicating chronic inflammatory responses that affect signaling in neural circuitry [195,202].

ASD children showed a significant change in the dynamics, α and β diversity, stability and relative abundance of GM when compared to healthy controls [19] and neuro-typical individuals [203]. GM of autistic children showed 10 times increased relative abundance and diversity of Clostridium spp. compared to healthy controls [204,205,206]. Two human gut microbiome studies showed increased numbers of three bacterial species (Desulfovibrio species, Bacteroides vulgatus, and Clostridia) in ASD children with GI symptoms compared to normal subjects with similar GI complaints [21,205]. Subsequent molecular-based studies showed a clear correlation between altered levels of Bifidobacterium, Lactobacillus, Sutterella, Prevotella, Ruminococcus, and Alcaligenaceae with autism [19,207,208]. ASD patients showed reduced Faecalibacterium (a beneficial species with anti-inflammatory and SCFA secreting functions) versus healthy controls [209]. Additionally, autistic children had reduced numbers of Akkermansia muciniphila, mucolytic bacteria, Prevotella and Bifidobacteria, and increased number of Alistipes, Lactobacillus, Bacteroides, and Prevotella [19,21,210,211] compared to controls. Taken together, these findings indicate that gastric disturbance, chronic gastritis and GD leads to a complex dysregulation of MGB signaling pathways which trigger a pathophysiological process that increases the occurrence of ASD.

### 4.5. ADHD

Attention deficit hyperactivity disorder (ADHD) is the most prevalent neurodevelopmental disorder, and is found mostly in children and adolescents, and characterized by triad symptoms like inattentiveness, impulsivity and hyperactivity [212,213]. Increased GI symptoms with altered microbiome composition are found in ADHD children. Four phyla—Proteobacteria, Firmicutes, Bacteroidetes, and Actinobacteria—are found to be dominant in ADHD patients. A clinical study showed that microbial α diversity was significantly reduced in ADHD patients compared with controls, while β diversity varied between patients and controls [124]. Another study demonstrated increased abundance of the Actinobacteria phylum, and reduced abundance of the Firmicutes phylum in ADHD patients compared to non-ADHD controls, and found a correlation between the abundance and the levels of hyperactivity and impulsivity [120]. Elevated levels of genus Bifidobacterium, and Bacteroidaceae and Neisseriaceae families were found to be associated with juvenile ADHD [120,124]. Overabundance of Bifidobacterium in the gut of ADHD patients correlated with significant increase in synthesis of cyclohexadienyl dehydratase that accounts for diminished neural reward anticipation [120]. In a cohort study, ADHD symptoms were found to be worse in ADHD children from mothers suffering from stress or infection than those children from unstressed mothers or healthy mothers [214]. Other studies found that ADHD children also had an increase in *Bacteriodes uniformis, Bacteriodes ovatus*, and *Sutterella stercoricanis*, and a decrease in *B. coprocola* species [122,208]. The presence of *B. ovatus* and *S. stercoricanis* abundantly showed a positive link with ADHD symptoms, and consumption of dairy products, nuts, legumes, seeds, ferritin, and magnesium correlate with *S. stercoricanis* abundance, while ingestion of fat and carbohydrate correlate with *B. uniformis* [122].

### 4.6. Schizophrenia

Schizophrenia (SZ) is a heterogeneous psychiatric disorder characterized by ‘positive symptoms’ (delusions and hallucinations) and ‘negative symptoms’ (apathy, anhedonia, fatigue, and lack of motivation) along with speech difficulty and cognitive decline [215,216,217]. However, the pathogenesis of SZ is still unknown, and a complex epigenetic mechanism (pathogenic association of genes with environmental factors) is believed to play a significant role [23,218]. Likewise, the onset age of SZ (range of 15–25 years) overlaps with the final stages of the pruning process at the synaptic regions of medial prefrontal cortex in adolescent brains [219]. Clinical studies have shown a direct association between GD and SZ, as the rates of GI symptoms reported in subjects with SZ are high [220]. In contrast to age-matched healthy control subjects, patients with first-episode psychosis had changed GM composition, with increased number of Lactobacillaceae, Halothiobacillaceae, Brucellaceae, and Micrococcineae and decreased number of Veillonellaceae. These changes had a positive correlation with the intensity of psychotic symptoms and the likelihood of remission after a year [221]. Furthermore, GD affects CNS by decreasing BDNF expression [55,222] in patients with SZ, and also by decreasing the level of NMDA receptor in the cortex and hippocampus of SZ patients [23,223].

In SZ patients, GD is marked by leaky gut syndrome caused by deregulated MGB signaling disturbances resulting in increased inflammatory markers [224]. Serum levels of C-reactive protein correlate with the severity of schizophrenic symptoms [225,226]. Evidence of MGB dysfunction includes histological findings like the presence of defective intestinal barrier wall, as colitis, enteritis and gastritis were diagnosed in 92%, 88% and 50% of autopsies report of 82 SZ patients, respectively [224,227]. Examples of immunological dysfunctions include increased levels of bacterial translocation markers like antibodies against Saccharomyces cerevisiae [228], bacterial markers like sCD14 and LPS binding protein [229], serum proinflammatory cytokines (mainly IL 1, IL 6, and IL 8) [230,231], and intestinal inflammation markers such as food antigen antibodies against cow milk and gluten content in wheat [228]. Meta-analysis of brain samples from SZ patients reveals significant reduction in the intracranial and total brain volume, especially in the grey matter of prefrontal cortex [232]. Neuropathological studies of SZ individuals [137,233,234] showed the presence of cortical thinning, more severely in frontal lobes. Recent research in SZ suggests the positive and strong pathogenic link between SZ onset and GD as immune-mediated inflammatory reaction affects the brain development, functioning of the nervous, immune, and endocrine system [23,235].

### 4.7. Bipolar Disorders

Bipolar disorders (BD) are chronic mood disorders, defined by the presence of prolonged depressive episodes and short manic or hypomanic episodes [236]. The chronic and persistent mood swings, cognitive deficits, and high irritability are the main clinical symptoms. Adolescent BD patients has shown significantly reduced grey volume in anterior and subgenual cingulate cortical regions [237]. A clinical study provided the causal link between the onset of manic episodes with irregular firing pattern in the ventral prefrontal-limbic networks due to dysregulated synaptic pruning process in BD patients [238]. PET measures confirm the correlation between the age of BD onset with the abnormal monoaminergic synaptic density due to altered synaptic pruning process [239]. Many clinical studies have shown reduced α-diversity in BD patients with respect to healthy controls [240,241] (Figure 3). GD-related BD is marked by reduced abundance of genus Faecalibacterium and family Ruminococcaceae and increased abundance of species *Faecalibacterium prausnitzii*, genera Bacteroides, Parabacteroides and Halomonas or Bacteroides–Prevotella group [127,242,243]. In addition, BD patients showed lower abundance of family Clostridiaceae [244] and butyrate producers such as genera Roseburia and Coprococcus [242], while showing increased abundance of genera Clostridium, Bifidobacterium, Oscillibacter, Streptococcus, Escherichia, Klebsiella, as well as Atopobium Cluster, Enterobacter species, Clostridium Cluster IV, phylum Actinobacteria, class Coriobacteria, and genus Flavonifractor [240,243].

The results reported in clinical studies and post mortem studies of NPD patients concur with the results of the preclinical studies. Moreover, GD-related pathological alterations in the systemic and nervous system vary depending on epigenetic and genetic factors, diet, and age of onset. A positive correlation between the deregulation of synaptic pruning process and onset of psychiatric symptoms with GD-induced MGB axis dysregulation was bolstered in several NPDs. Specific NPDs showed certain typical pattern of changes in diversity and composition. Anatomical alterations include grey matter reduction, cortical thinning, abnormal synaptic density, and abnormal firing pattern with defective neural circuitry. Reversal of GD using specific nutritional interventions or FMT have shown improvement of clinical symptoms and reduction of gastric symptoms.

## 5. Potential Microbial-Based Therapeutics in Neuropsychiatric Disorders

Alternative approaches targeting gut dysbiosis in NPDs are gaining much popularity after two decades of promising effects of GM and MGB on higher brain function, with enough evidence proving that significant effects of GD on anxiety and mood-related behaviors [246,247]. Recent studies have clearly shown the perturbations in the composition, diversity, stability, and maintenance of intestinal microbes negatively impact the host health and increase their vulnerability to a wide array of NPDs ranging from mild to severe form [36,64]. Several numbers of synthetic pharmaceutical agents are used to treat various NPDs in humans, although their effectiveness varies in clinical settings, and adverse effects often exceed the beneficial effects. A growing body of literature supports the preclinical results, and advises that reestablishing a healthy gut by developing well-balanced microbial composition and diversity (termed as eubiosis) through alternative approaches like fasting, calorie restriction, dietary modifications, and using supplements such as probiotics, prebiotics, synbiotics, or FMT [157,189,248]. Studies on the novel trigger-targeting therapeutics in the form of prebiotics, probiotics or synbiotics have reported clinical improvements in patients with neurological disorders [44,223]. Thus, these findings suggests that MGB axis signaling can be considered as a potential target in NPDs patients to improve the gastric and psychological symptoms.

### 5.1. Probiotics

Probiotic or probiotic mixtures consist of live microbes that, when ingested, stimulate the growth of beneficial microbial species [249]. The term “psychobiotics” generally denotes probiotics with higher neuroprotective properties used widely in the treatment of NPDs [3]. For example, treatment with lactic acid bacteria and Bifidobacteria has shown beneficial effects in patients suffering from multiple sclerosis, cognitive deficits, and stress-derived pathologies [144,250,251]. Lactobacillus and Bifidobacterium are the most common bacterial strains found to possess anxiolytic effects [189]. In animal models of anxiety, mixture of *Bifidobacterium (B.) longum + B. infantis + Lactobacillus (L.) helveticus + L. rhamnosus* alone or in combination with normalized behavioral phenotypes showed positive effects either by reversing the immune pro-inflammatory factors or upregulating the expression of GABA receptors [157,252,253]. Administration *of Lactobacillus rhamnosus* (*JB-1*) for prolonged periods reversed the altered GABA receptor levels in specific brain areas, and reduced corticosterone levels and depressive symptoms in mice [189]. *B. longum* supplementation reversed anxiety-like behavior and BDNF levels in brain samples of mice co-morbid with infectious colitis [157]. *L. farciminis* strains reduced leaky gut syndrome and HPA axis hyperactivity in rats exposed to stress [254]. A mixture of *L. helveticus* (strain R0052) + *B. longum* (strain R0175) prevented the reduction of neurogenesis in hippocampal regions of stressed mice [255]. Additionally, [193] showed that administration of Bifidobacterium strains reduced depressive- or stress-like behaviors. A probiotic mixture containing *L. helveticus and B. longum* showed anxiolytic-like activities in rats when exposed to physiological stress [256] and alleviated psychological distress in healthy human volunteers [246]. Previous reports on the beneficial effects of Lactobacillus and Bifidobacterium strains in alleviating GI symptoms of ASD children [144,189] was further confirmed by a recent study [257] in which a probiotic mixture of *L. reuteri* + *B. longum* increased the relative abundance of Lactobacillus. A mixture of Bifidobacteria, Streptococci and Lactobacilli significantly improved behavioral symptoms and reduced GI symptoms in ASD patients [258,259]. Probiotic supplementation with Lactobacillus rhamnosus GG in early life reduced the risk factors for ADHD development [260]. Bifidobacterium (*B. pseudocatenulatum* CECT 7765) intake promotes positive consequences with improved brain biochemistry and behavior in adulthood who were exposed to chronic stress in early life [261].

### 5.2. Prebiotics

Prebiotics are dietary ingredients that selectively promote the growth of beneficial microbes conferring health benefits to the host [262]. Studies have shown the beneficial effects of prebiotics in patients with depression, anxiety and cognitive deficits [263]. Prebiotics have also been reported to improve brain function, reduce disease symptoms and improve the overall wellbeing in patients with dementia [264], irritable bowel syndrome (IBS) [265], and ASD [266]. Dietary prebiotics such as non-digestible fructooligosaccharides (FOS) and bimuno galactooligosaccharides (BGOS) promoted the survival of useful species such as *B. longum* and reduced stress-induced activation of the HPA axis in healthy young participants [14,190]. A prebiotic mixture of FOS and galacto-oligosaccharide (GOS) improves depressive and anxiety symptoms in mice exposed to chronic mental stress [267]. BGOS administration reduced LPS-induced anxiety–depressive-like behavior in rats [268]. Prebiotic use leads to the amelioration or prevention of depression and anxiety by reducing plasma tryptophan and corticosterone levels as well as increasing the levels of 5-HT in the cecum [267,269,270]. BGOS administration for 3 months reduced anxiety symptoms and improved the quality of life in patients with irritable bowel syndrome [271], and improved cognitive flexibility in rats [272,273]. Short-chain FOS reduced anxiety and increased fecal Bifidobacteria abundance in IBS patients [274].

### 5.3. Synbiotics

Synbiotics refer to the synergistic combination of prebiotics and probiotics that confer benefits to the host by promoting the survival of beneficial microorganisms in the gut. Mild reduction in depressive symptoms and better cognitive functioning was found in aged people consuming synbiotic supplementation of FOS + probiotic for 24 weeks [275]. A randomized, double-blinded clinical study showed that synbiotic mixture of *Lactobacillus acidophilus* T16 + *Bifidobacterium bifidum* BIA-6 + *Bifidobacterium lactis* BIA-7 + *Bifidobacterium longum* BIA-8 reduced the negative symptoms in severely depressed patients along with higher serum levels of BDNF when compared to controls [276]. A synbiotic mixture of *Limosilactobacillus* (L.) reuteri + *Bifidobacterium* (B.) longum alone + GOS showed higher GI resistance, and improved the gut microbial activity and metabolism, marked by high levels of short-chain fatty acid and reduced levels of ammonium in autistic children [257]. Administration of synbiotics (*Lactobacillus paracasei* HII01 + *Bifidobacterium animalis* subsp. *Lactis +* galacto-oligosaccharides + oligofructose) in stressed individuals reduced the negative feelings, mainly by regulating the activation of HPA-axis and production of IL-10, IgA, and LPS [277]. A systematic literature review of clinical trials and observational studies in MDD patients showed that ymbiotic supplementation for 4 to 9 weeks moderately reduced depressive symptoms [278].

### 5.4. Dietary modifications

The main determinant of the composition and diversity of GM is diet (including dietary patterns, ingredients and composition). Beef-fed mice showed increased diversity of gut microbes, enhanced memory, and reduced anxiety compared to control mice (fed with normal chow) [279]. Population studies hinted at lower risk of depression and anxiety in subjects following “traditional dietary practice” [280]. Similarly, an Indian study [281] showed that healthy vegetarians have an increased abundance of Firmicutes (34%) compared to Bacteroidetes (15%), while healthy non-vegetarians have a reversed ratio of Bacteroidetes (84%) and Firmicutes (4%). Non-vegetarian diet elevated the relative abundance of Alistipes, Bilophila, and Bacteroides (i.e., bile-tolerant microorganisms) and reduced the relative abundance of Roseburia, Firmicutes Eubacterium rectale, and Ruminococcus bromii (i.e., microbes responsible for metabolizing plant polysaccharides [31]. In contrast, fiber-rich diets (derived from fruits, seeds, and vegetables) are found to be associated with healthy distribution of gut microbial composition. Dietary changes are also proposed as a therapeutic intervention for ASD symptoms. For example, ketogenic diet (primarily diet consisting of high fats, moderate proteins, and very low carbohydrates) showed marked improvement in behavior, eating habits, and tantrums in ASD children by regulating mitochondrial gene expression [282]. Ketogenic diet was shown to reduce seizure frequencies and improved behavioral symptoms such s learning abilities and social skills [283]. A 10-week randomized controlled study showed that micronutrient supplements reduced ADHD scores, correlating with higher abundance of Bifidobacterium [260]. Gluten-free diet enhanced behavior and augmented the circulatory levels of L-tryptophan [284], while ketogenic diet improved the clinical symptoms of SZ [285].

Latalova et al. (2017) [286] showed partial therapeutic effects of combined formulation of antibiotics and probiotics along with gluten + casein free diet in some SZ patients. A high-fiber diet was shown to lower colonic pH and prevented the overgrowth of pathogenic bacteria [287]. During early-life stressful episodes, omega-3 and omega-6 polyunsaturated fatty acids given as a supplementary diet protected the GM microbial composition and abundance, mainly by elevating the number of Bifidobacterium and Lactobacillus, and maintaining their metabolic activities [288,289].

### 5.5. Fecal Microbiota Transplantation (FMT)

A common method used in various NPDs is FMT, which includes transfer of complete or specific microbial components of a healthy donor to the recipient through endoscopy, enema, and oral feeding of frozen fecal materials in order to establish a healthy complex microbial composition and function (termed “eubiosis”) [290]. FMT treatment from healthy donors has shown beneficial effects in mice subjected to stressful conditions by alleviating depression- and anxiety-like symptoms. Transfer of fecal material from a healthy donor reduced depressive symptoms and anxiety in patients with irritable bowel syndrome and metal depression, and also reduced Clostridium difficile infection in elderly people [291]. Another study investigating the effects of FMT from depressed patients to antibiotic-treated animals (devoid of healthy GM) showed the development of anxiety-like behavior and reduced interests in normal pleasurable activities, mainly by altering tryptophan metabolism, compared with rats receiving FMT from healthy donor [292]. FMT is the most efficient and cost-effective approach in the treatment and management of NPDs, with zero or very few adverse effects.

## 6. Conclusions

Technical advancements and developments in recent times in the fields of neuroscience, neuroimmunology and gut microbiome research have provided ample evidence for the regulatory role of GM on the development, maturation, and function of the brain, as well as immune cells. Numerous GF studies using rodent models, antibiotic-treated animal studies, and transgenic animal models have specifically addressed the physiology of GM and gut-derived metabolites on the regulation of immune, metabolic, and brain functions via the MGB axis. These findings are further emphasized by the fecal transplantation experiments recently. Additionally, several animal experimental studies and observational clinical studies have demonstrated the correlation of GD with the development of complex mental disorders such as depression, ASD, SZ and ADHD. The common denominators of several NPDs include the presence of prolonged gastric symptoms, GD, increased inflammation (both at periphery and brain), altered microglial function, aberrated pruning process, abnormal neural circuitry and neuronal death in certain specific brain regions. Therefore, there is an urgent need to investigate the association between gut-derived microbial signals with microglial activation and neuroinflammation in order to effectively tailor the therapeutic process to obtain better efficacy and prevent disease progression in patients with NPDs.

Studies using microbial-based interventions have shown promising results in the improvement of patients’ health, as reversal of health gut microbiome improved the immune, endocrine and neural functions, which in turn reduced their psychiatric and gastric symptoms. Numerous studies in experimental animal models and humans testify to the potential benefits of special nutritional interventions such as prebiotics, probiotics and synbiotics, as well as dietary alterations and FMT in the treatment of wide array of NPDs by targeting GD. To develop personalized microbiota-based interventions in humans, better knowledge of the exact pathways underlying the MGB axis is mandatory. Specifically, immune-mediated inflammatory response is the central pathological feature of many mental disorders, and this can serve as better target for microbiota-based non-therapeutic approaches. Several plants-based supplements provide beneficial effects on establishing healthy GM and reducing the onset or progression of psychological disorders. Additionally, large longitudinal clinical studies investigating the pathogenic intestinal microbial changes in subjects with different NPDs are needed to discern their role in the pathophysiology of mental disorders.

## Figures and Tables

**Figure 1 cells-12-00054-f001:**
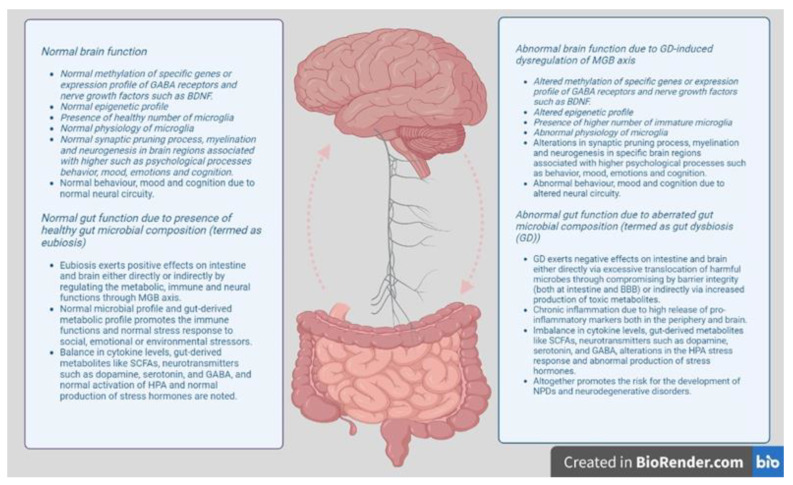
Gut–brain axis regulatory model. Healthy composition of gut microbiota (termed as eubiosis) maintains the integrity of the intestinal mucosa and mucus secretion, and regulates the endocrine, neural and immune signaling through the MGB axis. Healthy GM and GM-derived metabolic profiles regulate the immune function, the production of neurotransmitters and stress hormones, and maintains the function of glial cells, synaptic pruning and myelination. Thus, the MGB axis influences higher psychological functions such as mood, emotion, cognition and memory by regulating the neural circuits. Abnormal composition of gut microbiota (termed as dysbiosis) disrupts the intestinal membrane integrity and dysregulates the functions of endocrine, neural and immune system through MGB axis. Aberrated GM and GM-derived metabolic profile results in immune-driven inflammatory response, inadequate secretion of neurotransmitters, and increased stress response, immature and dysfunctional glial cells, impaired synaptic pruning, and myelination. Thus, deregulation of the MGB axis affects the higher psychological functions such as mood, emotion, cognition and memory by altered neural circuitry.

**Figure 2 cells-12-00054-f002:**
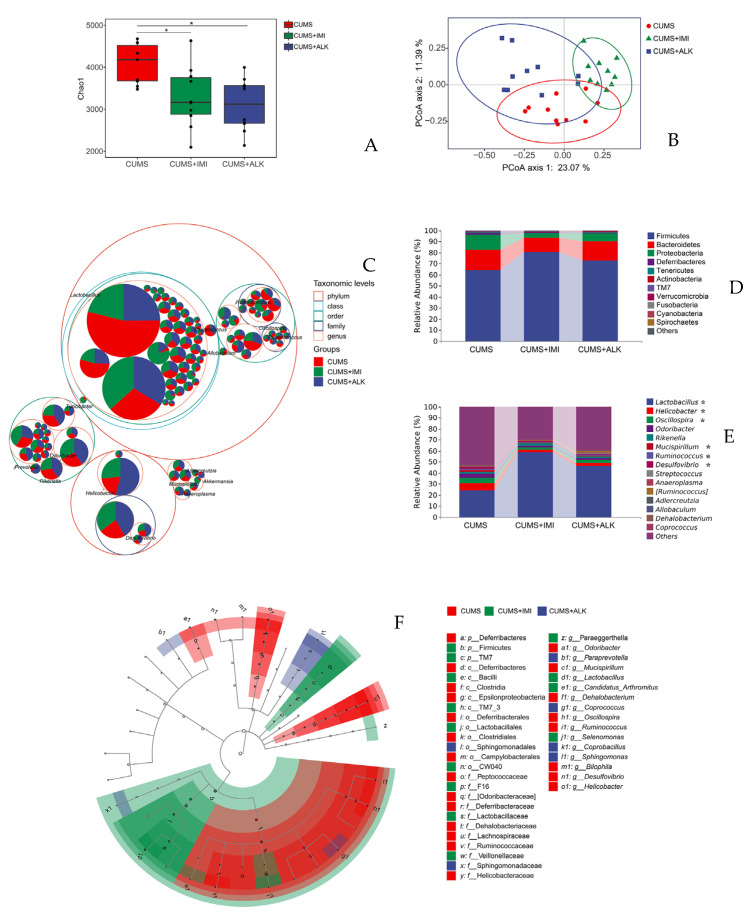
Composition and relative abundance of gut microbiota of mice subjected to chronic unpredictable mild stress (CUMS). The four groups are mice divided into no CUMS, CUMS (CUMS+ vehicle (0.9% saline)), CUMS mice treated with imipramine (CUMS + imipramine), and alkaloids (CUMS + alkaloids). (**A**) Chao1 index of imipramine and alkaloids-treated mice was reduced, but there was no significant difference in their diversity compared to saline treated CUMS mice. (**B**) The principal coordinates analysis (PCoA) based on the Bray–Curtis distance for GM showed a significant difference in microbial clusters among CUMS + imipramine and CUMS + alkaloids groups. Ellipses refer to 95% confidence intervals between treatments. (**C**) Taxonomic tree of GM represented in packed circles. The phylum level is denoted by the largest circles, while the inner circles are denoted by class, order, family, and genus. Firmicutes, Bacteroidetes, and Proteobacteria are found to be the dominant phyla. (**D**) Treatment with imipramine and alkaloids increased the Firmicutes, and decreased Bacteroidetes and Proteobacteria content. (**E**) Bar chart represents the specific changes in the relative abundance of GM between CUMS, CUMS + imipramine and CUMS + alkaloids groups at the genus level. * Denotes significant differences in specific microbes between groups. (**F**) Linear discriminant analysis scores on the cladograms of amplicon sequence variant is more than 2 based on linear discriminant analysis effect size [156].

**Figure 3 cells-12-00054-f003:**
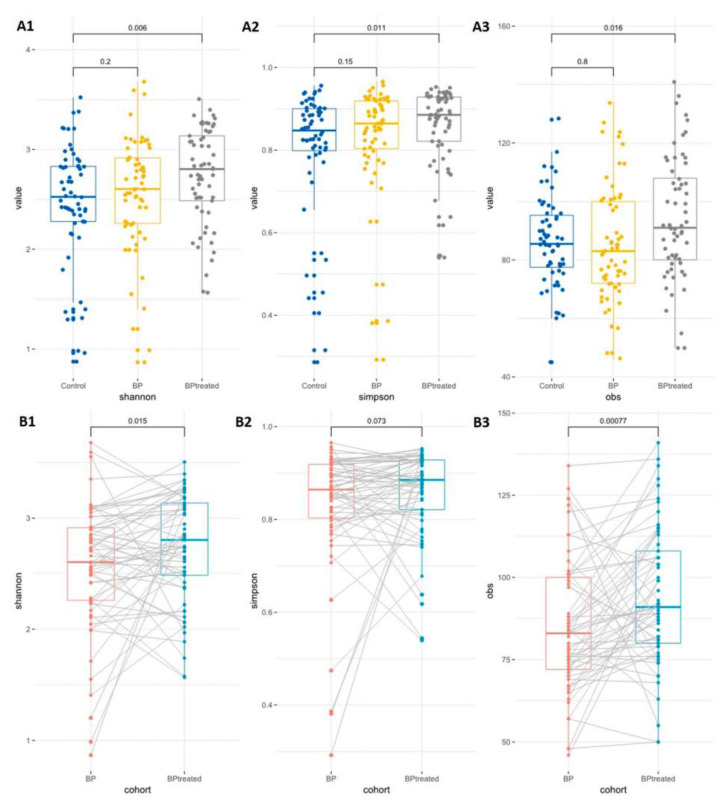
Gut microbiota diversity in healthy control and bipolar disorder subjects before and after treatment (Quetiapine, atypical antipsychotic medication). (**A**) α-diversity between control and untreated or treated subjects. (**B**) α-diversity between untreated and quetiapine treated subjects. As per the Shannon (**A1**), Simpson (**A2**) or obs (**A3**) indexes analyses, there is no significant difference in GM diversity between control and bipolar disorder subjects before treatment. However, treatment with quetiapine produced a significant difference in α-diversity between control and treated subjects. On the other hand, treatment with quetiapine improved the α-diversity in bipolar subjects when compared to the untreated subject. This effect was observed with the Shannon (**B1**) and obs (**B3**) indexes, but not in the Simpson index (**B2**) (the figure is reused as per journal copyright permission [245]).

## Data Availability

The data that support the findings of this study are available in standard research databases such as SCOPUS, PubMed, Medline, Cochrane, PsycINFO, Science Direct, or Google Scholar, and/or on public domains that can be searched with either key words or DOI numbers.

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
