# Peer review of "The Role of Gut Dysbiosis in the Pathophysiology of Neuropsychiatric Disorders"

_cells, 2022, doi:10.3390/cells12010054_

Round 1

Reviewer 1 Report

In this review, Nikhileshet al. discuss the role of gut dysbiosis in the pathophysiology of neuropsychiatric disorders. The manuscript is well-written and very interesting. I have only minor comments that can improve this review:

-          The manuscript should be proof-read for minor typographical errors.

-          Line 70-78: “Approximately 1013–1018”. Are those numbers correct? It should be exponentiation?

-          Throughout the manuscript, there are too many general statements not related to the focus of this review. The Authors should delete general information about pathologies ecc..

-          I suggest the Authors to discuss a recent research paper (PMID: 36400332), in which the Authors found a significant reduction of relative abundances of Actinobacteria and Verrucomicrobia in susceptible mice identified through an animal model of post-traumatic stress disorder (PMID: 33392367). These results are in line with the exploratory study cited at line 382.

Author Response

Reviewer -1: Comments and Suggestions for Authors

In this review, Nikhileshet al. discuss the role of gut dysbiosis in the pathophysiology of neuropsychiatric disorders. The manuscript is well-written and very interesting. I have only minor comments that can improve this review:

Comment 1 - The manuscript should be proof-read for minor typographical errors.

Response – We thank the reviewer for the comment. The revised manuscript is proof-read and rectified the typographical errors.

Comment 2 - Line 70-78: “Approximately 1013–1018”. Are those numbers correct? It should be exponentiation?

Response – It is exponentiation i.e 1013–1018. Appropriate corrections is made in the revised manuscript.

Comment 3 -   Throughout the manuscript, there are too many general statements not related to the focus of this review. The Authors should delete general information about pathologies etc.

Response – In this paper, pathological mechanisms are illustrated briefly to circumscribe wide array of neuropsychiatric disorders. Indeed, general content of pathologies is limited to a relevant level in the revised manuscript.

Comment 4 - I suggest the Authors to discuss a recent research paper (PMID: 36400332), in which the Authors found a significant reduction of relative abundances of Actinobacteria and Verrucomicrobia in susceptible mice identified through an animal model of post-traumatic stress disorder (PMID: 33392367). These results are in line with the exploratory study cited at line 382.

Response – We thank you for the suggesting the article (PMID: 36400332) related to our content. Results of the suggested research paper are added in the revised manuscript. We have also added results from another recent paper on PTSD-related alteration in gut microbiota written by Hoke A et al., 2022. (PMID: 35300376; PMCID: PMC8921487).

Reviewer 2 Report

The article showed the pieces of evidence on the potential interaction between gut dysbiosis and neuropsychiatric disorders through the preclinical and clinical data. Further, the author summarised the use of non-therapeutic modulators such as pro-, pre-, syn- and post-biotics, and specific diets or fecal microbiota transplantation promising targets for the management of neuropsychiatric disorders. However, there are a few issues that need to be considered and required correction:

Page 1; Line 26-27: It would be great if the statistically significant values should be used for the role of microbiota gut-brain axis in the abstract.

Page 2; Line 69: What methods author apply for manuscript construction? What is the evidence in terms of study types? Customization on the selection of the available pieces of evidence.

Required to add a section as Methods or data search immediately after the introduction which will justify the construction of the manuscript in a critical way. It should include the selection criteria such as the use of what kind of database sources used (Such as Scopus, MEDLINE, PubMed, Cochrane and ScienceDirect etc.), how did the filter occur (how many years of studies, and types of study including preclinical and clinical studies).

Further, each section of the review should be structured in a way the outcome of the studies should be written from the author's view, which is lacking in the whole manuscript.   

Author Response

Reviewer – 2: Comments and Suggestions for Authors

The article showed the pieces of evidence on the potential interaction between gut dysbiosis and neuropsychiatric disorders through the preclinical and clinical data. Further, the author summarised the use of non-therapeutic modulators such as pro-, pre-, syn- and post-biotics, and specific diets or fecal microbiota transplantation promising targets for the management of neuropsychiatric disorders. However, there are a few issues that need to be considered and required correction:

Comment 1 - Page 1; Line 26-27: It would be great if the statistically significant values should be used for the role of microbiota gut-brain axis in the abstract.

Response – Significant point on the role of MGB axis is added in the abstract of the revised manuscript.

Comment 2 - Page 2; Line 69: What methods author apply for manuscript construction? What is the evidence in terms of study types? Customization on the selection of the available pieces of evidence. Required to add a section as Methods or data search immediately after the introduction which will justify the construction of the manuscript in a critical way. It should include the selection criteria such as the use of what kind of database sources used (Such as Scopus, MEDLINE, PubMed, Cochrane and ScienceDirect etc.), how did the filter occur (how many years of studies, and types of study including preclinical and clinical studies).

Response – A new section of search methods and selection criteria are added in the revised manuscript.

Comment 3 - Further, each section of the review should be structured in a way the outcome of the studies should be written from the author's view, which is lacking in the whole manuscript.   

Response – The basic skeleton of the present manuscript is structured to provide an overall view of the gut microbes and its role in the onset or maintenance of neuropsychiatric symptoms. General outcome of the studies discussed under each section have been added in the revised manuscript. Conclusive findings are highlighted under major divisions.

Round 2

Reviewer 2 Report

The manuscript has improved from the previous version as addressed all the comments including the abstract, methods and author's view in each section of the manuscript.